# Thymosin α1 and Its Role in Viral Infectious Diseases: The Mechanism and Clinical Application

**DOI:** 10.3390/molecules28083539

**Published:** 2023-04-17

**Authors:** Nana Tao, Xie Xu, Yuyuan Ying, Shiyu Hu, Qingru Sun, Guiyuan Lv, Jianli Gao

**Affiliations:** 1School of Pharmaceutical Sciences, Zhejiang Chinese Medical University, Hangzhou 310053, China; 2State Key Laboratory of Quality Research in Chinese Medicine, University of Macau, Macao 999078, China

**Keywords:** thymosin α1, virus infection, immune regulation, protein structure, COVID-19

## Abstract

Thymosin α1 (Tα1) is an immunostimulatory peptide that is commonly used as an immune enhancer in viral infectious diseases such as hepatitis B, hepatitis C, and acquired immune deficiency syndrome (AIDS). Tα1 can influence the functions of immune cells, such as T cells, B cells, macrophages, and natural killer cells, by interacting with various Toll-like receptors (TLRs). Generally, Tα1 can bind to TLR3/4/9 and activate downstream IRF3 and NF-κB signal pathways, thus promoting the proliferation and activation of target immune cells. Moreover, TLR2 and TLR7 are also associated with Tα1. TLR2/NF-κB, TLR2/p38MAPK, or TLR7/MyD88 signaling pathways are activated by Tα1 to promote the production of various cytokines, thereby enhancing the innate and adaptive immune responses. At present, there are many reports on the clinical application and pharmacological research of Tα1, but there is no systematic review to analyze its exact clinical efficacy in these viral infectious diseases via its modulation of immune function. This review offers an overview and discussion of the characteristics of Tα1, its immunomodulatory properties, the molecular mechanisms underlying its therapeutic effects, and its clinical applications in antiviral therapy.

## 1. Introduction

The thymus is well-regarded as a vital primary lymphoid organ for its function in the generation of a vast variety of immunocompetent T lymphocytes [1]. The thymus, as the primary defense against pathogen invasion and a critical component of the immune system, is an important region for mounting immunological responses and sustaining immune function [2]. In particular, in the context of antiviral immunity, the thymus plays a critical and decisive role in directing the differentiation of CD8^+^ T cells, CD4^+^ T cells, Treg cells, and natural killer (NK) cells, thus serving as a critical factor in the host’s ability to mount effective immune defenses against a wide range of viral pathogens while maintaining immune system homoeostasis [3].

Thymosin α1 (Tα1) is a peptide hormone produced and released by thymic stromal cells that plays an important role in T cell differentiation, development, and maturation [4,5]. Thymosin was discovered and isolated in 1966 by Goldstein and colleagues, who found a lymphocyte production factor in the calf thymus that was later termed thymosin [6,7]. A protein combination including three short peptides (molecular weight less than 4000) known as thymosin fraction 5 (TF 5) was produced after separation and purification [8]. It was not until 1977 that Tα1 was purified and separated from TF 5 and found to possess 10–1000 times higher activity than TF 5 [9]. So far, researchers have found that Tα1 is a highly conserved peptide hormone that is extensively expressed in numerous mammalian organs, including the thymus, spleen, lung, kidney, brain, and blood, with the largest concentration in the thymus [10].

The immunomodulatory and immune-enhancing properties of Tα1 have long been recognized. Its synthetic derivative, thymalfasin, has been integrated into various clinical products and is now approved in over 35 countries for the treatment of hepatitis B and C, as well as for immune regulating effects in a variety of other disorders. Specifically, Tα1 has been used to treat a variety of illnesses, including acquired immune deficiency syndrome (AIDS), pseudomonas, and mold toxicity, as well as for treating sepsis. Most recently, Tα1 has been used in the treatment of severely ill patients with coronavirus disease 2019 (COVID-19).

Currently, the usefulness of Tα1 in treating viral infectious diseases, notably SARS-CoV-2 infection, is debatable. Therefore, relevant studies were systematically selected from PubMed (Medline), Scopus, and Web of Science databases up to 16 April 2023, with the aim of establishing a solid foundation for its clinical use for the treatment of viral infections. We hope that this review can provide a comprehensive summary and analysis of the characteristics of Tα1, including its structure and immunological regulation mechanism, as well as its clinical application.

## 2. Overview of Tα1

### 2.1. Properties of Tα1

Tα1 (generic drug name: thymalfasin; trade name: Zadaxin) is a bioactive peptide with 28 amino acid residues, which is obtained by cutting the front part of prothymosin α (ProT α, composed of 109 amino acid residues) by asparagine endopeptidase [11,12,13]. The sequences of the Tα1 peptide are as follows: Ac-Ser-Asp-Ala-Ala-Val-Asp-Thr-Ser-Ser-Glu-Ile-Thr-Thr-Lys-Asp-Leu-Lys-Glu-Lys-Lys-Glu-Val-Val-Glu-Glu-Ala-Glu-Asn-OH. Additionally, Tα1 has some interesting characteristics: (1) it has a relative molecular weight of 3108 Da [14]; (2) it is highly acidic with an isoelectric point of 4.2; (3) the N-terminal of Tα1 is acetylated, and there are no disulphide bonds or glycosylation structures [15,16,17]; (4) the entire polypeptide molecule has six amino acid repeats (Ala-Ala, Ser-Ser, Thr-Thr, Lys-Lys, Val-Val, Glu-Glu); (5) it may be involved in the regulation of the cell cycle [18]; and (6) it can indirectly affect transcription and/or DNA replication [19].

### 2.2. The Structure of Tα1

Under natural conditions, Tα1 is a short, highly charged, and inherently disordered protein. At neutral pH and 37 °C, Tα1 typically exhibits intrinsic disorder, meaning that it does not have a stable, defined structure [16]. In the monolayer vesicle of dimiristoylphosphatidylcholine and dimiristoylphosphatidic acid (10:1), it showed a partially structured conformation [20]. Under low pH conditions, Tα1 has the ability to build ordered protein complexes through interaction with other naturally existing proteins. Moreover, in organic solvents such as trifluoroethanol, hexafluoropropanol, or sodium dodecyl sulfate (SDS), Tα1 is commonly observed to adopt a stable conformation. A structured conformation of the peptide was observed through restrained molecular dynamic simulations with an explicit solvent comprising 40% TFE/60% TIP3P water (*v*/*v*), with two stable regions identified: an alpha-helix region spanning residues 14 to 26, and two double β-turns in the N-terminal twelve residues of Tα1, which form a distorted helical structure (Figure 1a) [15]. Additionally, two β-rotational conformations were detected at the N-terminal of Tα1, namely (I, I + 1)-double rotations of the residues ASP2-ASP6 and (I, I + 2)-double rotations of the residues Thr7-Thr12. In contrast to TFE mixed solvents, Tα1 in SDS displays a spiral folded conformation (Figure 1b) [21]. This conformation is characterized by a structural fracture between residues 1–9 and 14–25, with the acetylated N-terminal residues 1–5 of Tα1 often inserted into the hydrophobic region of the micelle. The investigation also revealed that the folded conformation of Tα1 in SDS closely resembled that in phospholipid vesicles, taking on a 3_10_ helical structure [21]. Nonetheless, a few differences were observed in the tertiary structure of Tα1 between these two environments (Figure 1c).

As a result, when Tα1 is folded on a membrane with negative charge exposed on the surface, it may connect with receptors on or near the membrane and insert the N-terminal of Tα1 into the hydrophobic area of the membrane, resulting in a bio-signaling cascade response [21]. For instance, the activation of the phosphorylation pathway of I(κ)B kinase (IKK) via the tumor necrosis factor (TNF) receptor-related factor 6 (TRAF6) can be induced by Tα1 [22]. Nevertheless, due to the limitations of full peptide encapsulation, the potential interaction between Tα1 and the cell membrane remains unknown.

### 2.3. The Protein Binding Properties and Biosafety of Tα1

Despite the pleiotropic effects of Tα1 on immune regulation, the lack of particular receptors remains one of the fundamental factors leading to the inefficacy of therapy with Tα1. However, recent research has demonstrated that the C-terminal portion (residues 11–20) of Tα1, which is defined by the amino acid sequence “LKEKK”, is capable of binding to human serum albumin (HSA) [23]. HSA is a serum protein that can serve as a carrier for a variety of medicines and polypeptides. The C-terminal sequence of Tα1 can be delivered to the vicinity of a target membrane exposing phosphatidylserine (PS) through the assistance of HSA, under conditions where the membrane region is negatively charged. The N-terminal region of Tα1 can then enter into the hydrophobic region of the cell membrane, producing a cascade reaction of biological signals (Figure 2) [17,23]. As a result, plasma proteins may act as carriers of Tα1 targeting areas. However, the combination of the two is not close; it just creates conditions for the binding and diffusion of Tα1.

Furthermore, electrostatic interactions may enhance the binding of Tα1 to hyaluronic acid (HA) and interfere with the binding of HA to CD44 and the motor receptor RHAMM, inhibiting viral infection progression [24]. HA is a glycosaminoglycan found on the cell surface and in extracellular media that interacts with RHAMM and CD44 via a shared BX7B motif, where “B” is an Arg or Lys residue and “X” is any amino acid with no basic properties. Nevertheless, no definitive BX7B motif is found in the amino acid sequence of Tα1. When the sequence of Tα1 was compared to the HA receptor sequence, it was discovered that the C-terminal portion of Tα1, specifically at the lysine residue position in the “LKEKK” sequence, shows sequence resemblance to the HA receptor. Specifically, the sequence of CD44 at positions 41–45, 153–162, and 711–719, as well as the sequence of RHAMM at positions 743–750 and 721–731, all exhibit similarities with the sequence of Tα1. The positively charged lysine residue side chains on Tα1 may create ion bridges with negatively charged HA, thereby potentially interfering with the binding of HA to certain receptors such as CD44 or RHAMM and their complicated interactions.

Tα1 has garnered substantial clinical therapeutic attention due to its various biological effects. There are currently three primary approaches for the production and purification of Tα1: biological extraction, chemical solid-phase synthesis, and gene engineering expression [25]. Solid-phase synthesis is the only technique approved for the clinical production of Tα1. Tα1 is commonly provided twice a week through subcutaneous injection, with a conventional dosage range of 0.8–6.4 mg and a multi-dose range of 1.6–16 mg [26]. A pharmacokinetic study showed that after subcutaneous injection, Tα1 is well-absorbed in the body, and its peak blood drug concentration (Cmax, the highest blood drug concentration after administration) is reached at 1–2 h, with a plasma half-life (t_1/2_, an estimate of the time it takes for the concentration or amount in the body of that drug to be reduced by exactly one-half (50%)) of less than 3 h [27]. Tα1 usually has good security. The most common adverse reactions include local irritation, redness, or discomfort at the injection site. However, Tα1 is often banned in immunocompromised individuals due to its immunomodulatory action (such as organ transplant patients).

### 2.4. The Immunomodulatory Mechanism of Tα1

Tα1 is a well-known polypeptide with immunoregulatory effects as well as biochemical features [28]. Tα1 has shown encouraging outcomes in viral infectious disorders such as hepatitis B, either alone or in combination with other medications. The direct effect of Tα1 on lymphoid cells might explain some of the reported effects. Tα1 exerts an immune modulatory activity on T cell and NK cells, and impacts the functions of mature lymphocytes, such as stimulating cytokine production and cytotoxic T-lymphocyte-mediated cytotoxic responses [29]. Presently, the non-specific immune regulation mechanisms of Tα1 are now classified as direct and indirect immunological processes (Figure 3) [30].

#### 2.4.1. The Effects of Tα1 on Immune Cell

Tα1 increases the expression of major histocompatibility complex (MHC) antigens and B-2 microglobulin on the cell surface, resulting in increased expression of virus-specific antigens and a reduction in viral replication [19,31,32,33]. A previous study has shown that Tα1 can increase the expression level of glutathione [34]. Simultaneously, one study discovered a negative association between glutathione concentration and influenza virus replication, indicating that glutathione may play an essential role in inhibiting influenza virus replication [35]. Tα1 may indirectly decrease virus multiplication and increase immune response by raising glutathione content via glutathione-dependent antiviral mechanism. However, further research is needed to confirm this possibility. Studies conducted in vitro have indicated that Tα1 can significantly impact T cell production and maturation. Additionally, Tα1 can stimulate the production of cytokines in T-helper 1 (Th1) cells, such as interferon-γ (IFN-γ) and interleukin-2 (IL-2), and activate NK cell-mediated cytotoxicity [36]. In cancer and cyclophosphamide immuno-suppressed mice, Chen et al. suggested that Tα1 administration can increase NK cell activity [37]. Furthermore, by stimulating NK cells, Tα1 can enhance the body’s ability to eliminate virus-infected cells. Tα1 has also accelerated the recovery rate of NK activity in bone marrow-reconstituted murine chimeras [38].

Dendritic cells (DCs) are powerful antigen-presenting cells (APCs) that play an important role in the immune response [39]. Tα1 has activated subsets of myeloid DCs (mDCs) and plasmacytoid DCs (pDCs). Both immature and mature DC subsets are capable of phagocytosing conidia. Tα1 enhances the phagocytic activity of immature DCs, modifies the morphology of DCs, and increases the expression of HLA class Ⅱ antigens and costimulatory molecules in response to conidia. Regarding cytokine production, it has been demonstrated that Tα1 significantly promotes the release of IL-12 p70 by immature mDCs in response to conidia and zymosan, and increases the production of IL-10 by immature pDCs in response to conidia [40]. Interestingly, DCs are important not only in eliciting immunological responses but also in promoting immune tolerance [19]. Tα1 can activate TLR9, induce the expression of indoleamine 2,3-dioxygenase (IDO) in DCs, and then activate the tryptophan catabolism-induced immune suppression pathway in vivo [41,42]. As a result, the production of IL-10 in CD4^+^CD25^+^ regulatory T cells is stimulated [43].

It is evident that Tα1 can affect the maturation, differentiation, and function of T cells. Recent research has also found that DC subsets have a significant polarizing impact on T-helper differentiation. Tumor necrosis factor-alpha (TNF-α) is one of the substances capable of stimulating DC maturation and IL-12 production in vitro. Huang et al. demonstrated that, during the maturation of bone marrow-derived DCs (BMDCs), Tα1 promoted the differentiation of CD4-expressing DCs and the expression of activation markers, but without affecting the production of IL-12, as well as the T cell-stimulatory capacity of DCs in the absence of TNF-α [39]. However, in the presence of TNF-α, Tα1 has been shown to not only raise the expression of CD4 on MHC class Ⅱ^+^ DCs and boost the up-regulation of mature markers caused by TNF-α, but also to decrease the up-regulation of IL-12 production. These effects were most noticeable at the therapeutic doses of Tα1.

Furthermore, Tα1 can affect immune function by inducing and regulating the maturation of T and NK cells, activating lymphocytes, and regulating the secretion of inflammatory cytokines such as IL-2, IL-4, IFN-γ, TNF-α, et al. [41,44,45]. Upon binding to TLR receptors located on the surface of precursor T cells, Tα1 promotes their differentiation into cytotoxic T lymphocytes (CD8^+^ T cells, CTL) [29,31,46]. These CTLs can recognize damaged or low-expression MHC-Ⅰ cells and trigger the release of IFN-γ, thereby controlling viral replication [47]. In conjunction with NK cells, CTLs form an integral defense line of antiviral immunity.

CD4^+^ T-helper2 (Th2) cells are important in immune response regulation because they activate T-dependent B cells and promote the generation of virus-specific antibodies. It should be noted that CD4^+^ T cells are more susceptible to viral infections than other types of immune cells [48]. By increasing the number of CD4^+^ T cells and effectively maintaining the CD4^+^/CD8^+^ T cell ratio, Tα1 is capable of exerting a significant positive impact on the immune system of the host organism [49].

The monocytic/granulocytic system (including the differentiated macrophages) and the principal cellular effectors of the immune response, play a crucial role in identifying and eliminating foreign entities such as pathogenic microorganisms [50]. Research by Peng et al. demonstrated that Tα1, as a weak immune modulator, can directly activate bone marrow-derived macrophages (BMDMs) to produce IL-6, IL-10, and IL-12 [51]. Moreover, Tα1 promptly stimulates the assembly and disassembly of podosomal structures, thus affecting the motility, invasion, and chemotaxis of BMDMs [52].

#### 2.4.2. The Effects of Tα1 on Inflammation Related Signaling Pathways

Tα1 is also the major activator of Toll-like receptors (TLRs) in myeloid and plasma cell-like DCs. DCs, a type of mononuclear phagocyte, are often regarded as the most efficient antigen-presenting cells and play a critical role in modulating both innate and adaptive immune responses [53]. TLRs, which belong to the class Ⅰ transmembrane receptor family, are present on the cell membrane surface or expressed on organelle membranes. The typical signaling pathways for TLRs include myeloid differentiation factor 88 (MyD88), IL-1 receptor-related kinase activator (IRAK), and TRAF6 [54]. MyD88 serves as the key adaptor protein that triggers the activation of Nuclear Factor Kappa B (NF-κB) in the signaling cascade induced by Tα1. Tα1 can directly induce functional maturation of DCs via TLRs (such as TLR2, TLR3, TLR5, TLR9, et al.), activate signal transduction pathways, such as the MyD88-dependent pathway and the p42/44 mitogen-activated protein kinase (MAPK)/c-Jun NH2 terminal kinase (JNK) pathway, and enhance the secretion of cytokines, such as IL-6, IL-10, IL-12, IL-13, and IL-17, thereby conferring protection against viral infections [40,55,56]. Studies have indicated that Tα1 can induce the expression of IL-6 through the TRAF6/atypical protein kinase C (PKC)/IKK/NF-κB pathway [22].

Tα1 can stimulate the expression of IL-6, IL-10, and IL-12 by activating the IRAK4/1/TRAF6/PKCζ/IKK/NF-κB and TRAF6/MAPK/AP-1 signaling pathways [51]. Furthermore, the p38 MAPK/NF-κB and TLR9/MyD88/IRF7 pathways are also potential mechanisms by which Tα1 activates DCs, inducing IFN-α/IFN-γ-dependent pathways and antiviral responses in vivo [57,58,59]. Moreover, Sodhi et al. suggested the activation of the p42/44 MAPK/JNK pathways in response to in vitro treatment with Tα1 in murine BMDMs [55]. The maximal expression of phospho-p42/44 MAPK was observed after 5–15 min following stimulation with 100 ng/mL of Tα1. Moreover, Tα1 can activate a TRAF6-atypical PKC-IκB kinase signaling pathway that activates NF-κB, which in turn triggers cytokine gene expression in murine BMDMs [60].

In summary, the antiviral effect of Tα1 can be summarized into two aspects: on the one hand, Tα1 can directly inhibit virus replication and viral protein expression by increasing the expression of cell surface-related antigens [19,31,32,33]; on the other hand, after the virus enters the body, Tα1 can treat viral diseases by enhancing T cell function, activating dendritic cells and macrophages, increasing the phagocytic activity of dendritic cells and the cytotoxicity of NK cells, activating TLRs, and starting MAPK, Jak, NF-κB, and other signaling pathways [22,51,54,55,57,58,59].

## 3. The Application of Tα1 in Viral Diseases and Its Complications

Tα1 has shown significant effectiveness as an immunomodulatory drug in the treatment of various viral infections. It has been used in clinical settings for almost three decades to treat a wide range of viral infectious disorders, including sepsis, hepatitis B, and others.

### 3.1. COVID-19

Coronaviruses are RNA viruses with a capsule and a linear, single-stranded, plus-stranded genome that only infect vertebrates. Their genome ranges in size from 26 to 32 kb [61]. The emergence of SARS-CoV-2 and its associated disease, COVID-19, in 2019 led to a global public health crisis of unprecedented magnitude. As of 13 April 2023, the virus had infected over 762 million people worldwide, resulting in over 6.8 million deaths “https://coronavirus.jhu.edu/ (accessed on 13 April 2023)”. Early research has linked elevated blood levels of pro-inflammatory cytokines (such as IL-1, IL-6, IL-8, IL-12, IFN-γ, IP-10, and MCP-1) to lung inflammation and damage in SARS patients [62]. The levels of cytokines (such as TNF-α, IFN-γ, MCP-1, IL-8, IL-10 and IL-1β) in patients were higher than those in healthy adults [63]. Huang and others have also shown that patients with COVID-19 had a high level of inflammatory cytokines (such as IL-1β, IFN-γ, IP-10, and MCP-1). The increase in the concentration of inflammatory cytokines not only activates the Th1 cell response, but also promotes the Th2 cell response in vivo, increasing the amount of inflammatory suppressors [64]. As a result, an exaggerated inflammatory response has also become a major marker of infection with COVID-19 [65]. In addition, infection with COVID-19 might alter antigen presentation function or the pool of tissue-resident immune cells, leading to the inability of the body to produce a prompt and efficient antiviral response, thus reducing the virus clearance rates and causing tissue damage [66,67].

Tα1 is often used as an immunopotentiator in clinical applications for viral infections (e.g., hepatitis C, AIDS). Because delayed immune reconstitution and the cytokine storm are still major barriers to the therapy of COVID-19, several researchers have begun to investigate Tα1 for the treatment of COVID-19 infection [65]. It has been shown that therapy with Tα1 lowers the RNA shedding time in COVID-19 and improves the viral clearance rate [68]. In addition, several data sets from in vitro studies have also demonstrated that Tα1 treatment helps alleviate lymphopenia in patients with COVID-19 and can promote the proliferation and differentiation of effector T cells, with the potential to regulate immune homoeostasis and cytokine storm in vivo [68,69]. Tα1 has been found in several trials to dramatically reduce mortality within 28 days of critical illness by increasing the oxygenation index (i.e., PaO_2_/FiO_2_ ratio) [70]. For patients whose CD8^+^ T cells or CD4^+^ T cells are lower than 400/μL or 650/μL, respectively, Tα1 can effectively increase the number of T cells in the blood of patients with severe lymphopenia [71]. Matteucci et al. cultured the blood cells of 15 patients in vitro [67]. This study found that Tα1 can reduce the expression level of cytokines in peripheral blood of patients with COVID-19. Tα1 can also regulate 39 genes in CD8^+^ T cells. The percentages of CD38, intracellular IL-6, and IFN-γ in the subsets of CD8^+^ T cells following treatment with Tα1 varied according to illness severity. Current research has identified eight primary biological targets (including HLA-A, Flynn protease, DPP4, CTSL, CTSB, ACE, PTGS2, and HLA-DRB1) for the treatment of COVID-19 [71]. Tα1 significantly binds to angiotensin converting enzyme (ACE) at extremely low concentrations, effectively downregulating the expression of ACE 2 and impairing the synthesis of angiotensin in human lung epithelial cells [72]. These results indicate that Tα1 may prevent COVID-19 by reducing the expression of ACE 2 in human lung epithelial cells [72,73]. However, some experiments have proved that Tα1 is not always effective in treating COVID-19. Liu et al. showed that the unhealed rate of patients in the treatment process was related to the use of Tα1, and patients who started using it later had a relatively high unhealed rate [74]. Wang et al. showed that Tα1 had no positive effect on the recovery of CD4^+^ and CD8^+^ T cell levels in patients [71]. At the same time, the duration of COVID-19’s RNA shedding in the upper respiratory tract of patients treated with Tα1 was significantly longer than that of patients in the non-Tα1 treatment group. Sun et al. included 771 patients in their study, and the overall death rate was 60.6%, whereas in the Tα1 treatment group, it was 41.3% (*p* < 0.001) [75]. However, after controlling for the confounding factors at the outset, there was no significant link between the treatment of Tα1 with a reduction in 28-day mortality, and no significant difference in mortality across groups (Table 1).

In a word, the safety and efficacy of Tα1 in patients with COVID-19 are uncertain. One explanation for this is that Tα1 has yet to be validated in large-scale randomized controlled studies. Although some small-scale studies show that Tα1 may be advantageous to immunological adjuvant treatment for patients with COVID-19, the sample size is minimal, and the study techniques are flawed. Therefore, more rigorous and standardized research is required to assess the safety and efficacy of Tα1 in patients with COVID-19. Another reason is that the immune response mechanism caused by COVID-19 may be different from other infections. Siddiqi et al. showed that the immunological response to the infection of COVID-19 is separated into three stages: stage Ⅰ is an asymptomatic incubation phase; the body usually shows no further abnormalities except for lymphopenia and neutropenia. Antiviral treatment is advantageous. Stage Ⅱ is the non-severe symptomatic phase; the patient’s inflammatory levels begin to rise, and antiviral medication should be taken with caution and reason. Stage Ⅲ is the severe respiratory symptomatic stage; patients’ inflammatory markers such as IL-6, IL-8 and TNF-α rise, making them prone to a “cytokine storm” response [79,80]. Most clinical data appear to demonstrate that as the virus infection progresses to the third stage, antiviral immunotherapy no longer has a favorable impact. Therefore, it is necessary to carefully consider when to use Tα1 to avoid causing excessive immune response. Moreover, additional studies are required to define the application breadth and the optimal treatment regimen for Tα1. The manner of usage, dose, and course of treatment of the intervention therapy with Tα1 may need to be changed individually for individuals with COVID-19 of varying severity. As Lin et al. highlighted, a rise in complement levels indicates that the body has launched an inflammatory reaction [77]. When the clinical patients treated with Tα1 are reviewed, the level of complement C3 in patients with COVID-19 treated with Tα1 is shown to be considerably lower than in the control group. The conclusion is that Tα1 therapy has a positive effect. To summarize, to reduce the controversy of Tα1 on the diagnosis and treatment results of patients with COVID-19, we should pay full attention to the selection and use of effectiveness markers in clinical studies. Although CRP, PCT, IL-6, and other variables are usually used as biomarkers for differential diagnosis and prognosis prediction of anti-infective immune response, more clinical data are still needed to support them as the diagnosis and treatment criteria for COVD-19.

### 3.2. Other Viral Diseases and Its Complications

#### 3.2.1. Hepatitis B

Hepatitis B virus (HBV) is the main cause of chronic liver disease, and more than 257 million people worldwide suffer from chronic hepatitis B (CHB). The persistent existence of viruses with covalently closed cyclic deoxyribonucleic acid (cccDNA) makes the immune system unable to establish an adequate immune response [81]. As an antiviral agent, the clinical efficacy of Tα1 in treating HBV infection has been confirmed by many studies [82,83] (Table 2).

#### 3.2.2. Hepatitis C

Chronic infection with the hepatitis C virus (HCV) has long been recognized as a global public health issue, eventually leading to the development of cirrhosis, end-stage liver disease, and HCC. Nevertheless, employing Tα1 alone to treat hepatitis C patients did not produce the desired results [84]. Clinical research showed that the therapeutic efficacy of Tα1 coupled with IFN-α is obviously better than that of IFN-α alone. The key assessment measure in this study was the sustained viral response (SVR) rate of patients after taking treatment. Among 552 eligible cases, 275 patients were randomly selected as the experimental group, and received PEG-IFN-α-2a (180 μg/week) + RBV (1000–1200 mg/day) + Tα1 (1.6 mg, twice a week), whereas 277 patients in the control group received PEG-IFN-α-2a (180 μg/week). The experimental results showed that the SVR rate of the Tα1 combined treatment group was significantly higher than that of the control group (41.0% vs. 26.3%, *p* < 0.05), and there was no significant difference in the incidence of adverse events among different groups, which confirmed the relative safety of Tα1 [85].

#### 3.2.3. AIDS

AIDS is an immunodeficiency disorder caused by infection with the human immunodeficiency virus (HIV). It primarily targets immunological cells that can express CD4, such as DCs, macrophages, and CD4^+^ T cells [86]. According to one research, combining Tα1, zidovudine and IFN-α may successfully enhance the number of CD4^+^ T cells in patients with AIDS [87]. Unfortunately, there is little data to prove that Tα1 can effectively control HIV infection as a monotherapy, hence more research is needed to confirm its efficacy and safety.

#### 3.2.4. Sepsis

Viral infection often causes an inflammatory reaction and affects the normal function of the immune system, which leads to the decline of the body’s immunity and increases the risk of infection with bacteria and other pathogens. Sepsis is an organ dysfunction syndrome caused by the imbalance of human response to infection [88,89]. Its main characteristic is systemic inflammatory response syndrome (SIRS) triggered by an infection [90]. According to Chen et al., the combined administration of Tα1 and Xuebijing has been shown to improve the therapy effect of severe pneumonia complicated with sepsis, improve the hemorheology condition of patients, and reduce the expression levels of serum CRP, TNF-α, IL-6, IL-8, and other inflammatory factors in patients [91].
molecules-28-03539-t002_Table 2Table 2Clinical research of Tα1 on other diseases.DiseaseDosage RegimenSubjects with Tα1/SubjectsMain ResultsPublished YearHBVTα1+entecavir345/690▪Combination therapy with Tα1 tends to inhibit the development of HCC2018 [33]Tα1228/468▪The _Δ_NLR level was considerably lower in those who received Tα1 postoperatively compared with those who did not2021 [92]Tα1146/558▪Reduce the recurrence rate of HCC▪Improve liver function index of patients2016 [93]Tα1+PEG-IFN-α-2a26/51▪The short-term addition of Tα1 was not superior to PEG-IFN-α-2a2012 [94]Tα125/40▪Improve the level of cytokines, especially IFN-γ2010 [95]Tα1+lavmidine34/67▪The combination treatment of Talpha1 and lamivudine did not have an obvious benefit of virological and biochemical response2008 [96]HCVTα112/24▪Promotes a prompt activation of NK compartment2010 [38]Tα1+IFN22/41▪May enhance the end of treatment response in naive patients with chronic hepatitis C2003 [97]AIDSTα16/12▪Without significant toxicities▪No additional increase in CD4^+^ T cell count1996 [98]Tα113/20▪PBMC sjTREC levels increased▪May enhance immune reconstitution2003 [99]

## 4. Conclusions & Future Perspective

As an immunomodulatory agent, Tα1 has the following characteristics. To begin with, Tα1 has excellent biological safety. There have never been any adverse reports of drug–drug interactions in prior studies of this drug’s clinical use. It was employed as an immune adjuvant in numerous viral infectious illnesses in clinical trials (such as HBV, immunodeficiency diseases, et al.). For example, in 2003, Tα1 was used as an immunopotentiator for patients with SARS, which effectively controlled the progress of the patients with SARS.

In addition, Tα1 demonstrates favorable pleiotropic functions as it can elicit various immunomodulatory effects based on the immune status of the afflicted individual. As a polypeptide, Tα1 possesses high selectivity and thus serves as a promising lead compound [100]. Additionally, Tα1 can be modified synthetically, thereby improving its stability and bioavailability regardless of plasma stability and oral bioavailability [100]. This advantageous feature of Tα1 makes it an attractive candidate for use in immunotherapy strategies.

In summary, Tα1 promotes the immune function of DCs, macrophages, NK cells, and T cells. It achieves this by enhancing the phagocytosis and antiviral ability of DC, activating Toll-like receptors on its surface, and activating signal pathways such as MAPK, Jak, and NF-κB, releasing cytokines such as interleukin and interferon. Although the effectiveness of Tα1 in treating patients with COVID-19 remains a topic of controversy, our review suggests that early administration of Tα1 in patients with COVID-19, rather than those in the late stage (cytokine storm), may yield better clinical outcomes. Notably, Tα1 was effective in controlling the progression of patients with SARS in 2003 as an immune enhancer. We anticipate that with further research and understanding of the COVID-19 disease course and the treatment of Tα1, a safe and effective treatment protocol for Tα1 can be developed for patients with COVID-19.

## Figures and Tables

**Figure 1 molecules-28-03539-f001:**
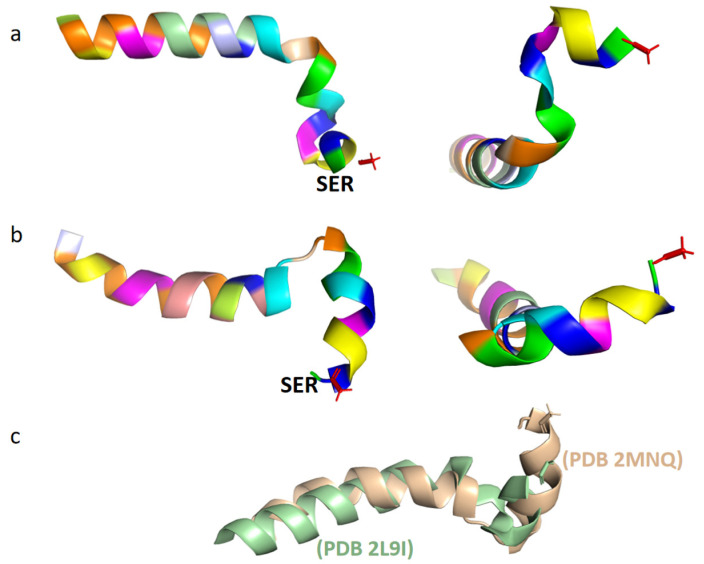
Tertiary structure of Tα1 in different solvents (analyzed and processed by the software of PyMOL 2.4.1). (**a**). Spatial conformation of Tα1 in a TFE mixed solvent (PDB 2L9I; Different colors represent different amino acids.). (**b**). Spatial conformation of Tα1 in an SDS micellar solvent (PDB 2MNQ; Different colors represent different amino acids.). (**c**). Details of the structural differences between Tα1 (in green) and SDS micelles (in brown) in a mixed solvent (obtained through superposition comparison of protein structures using the software of PyMOL 2.4.1).

**Figure 2 molecules-28-03539-f002:**
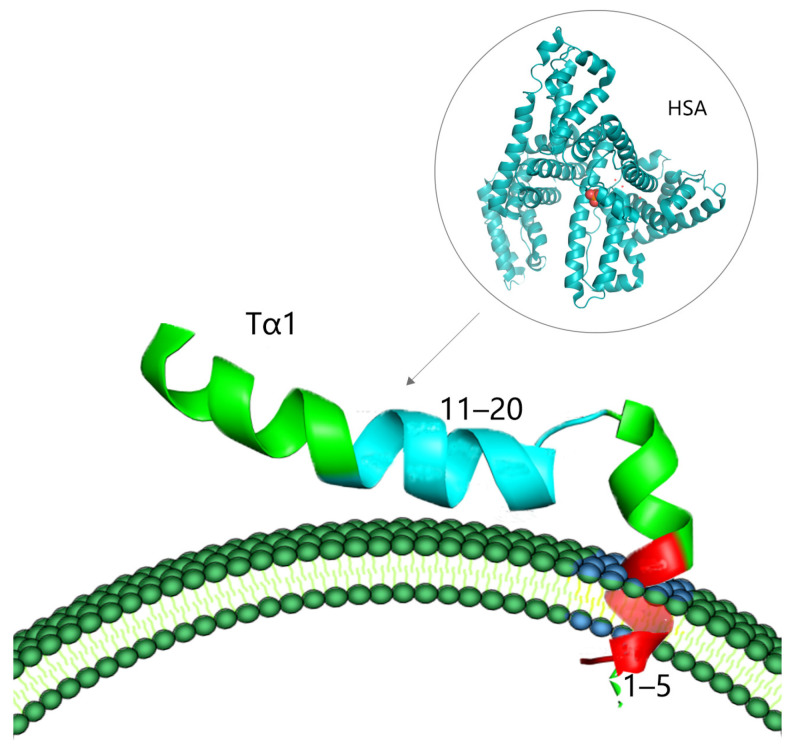
Possible interactions between Tα1 and the membrane (the dark blue part of the phospholipid bilayer is the negatively charged target membrane region after PS exposure, which is combined with the 1–5 amino acid residue sequence of the red part of Tα1; the 11–20 amino acid residue sequence of the blue part of Tα1 is the region interacting with HSA).

**Figure 3 molecules-28-03539-f003:**
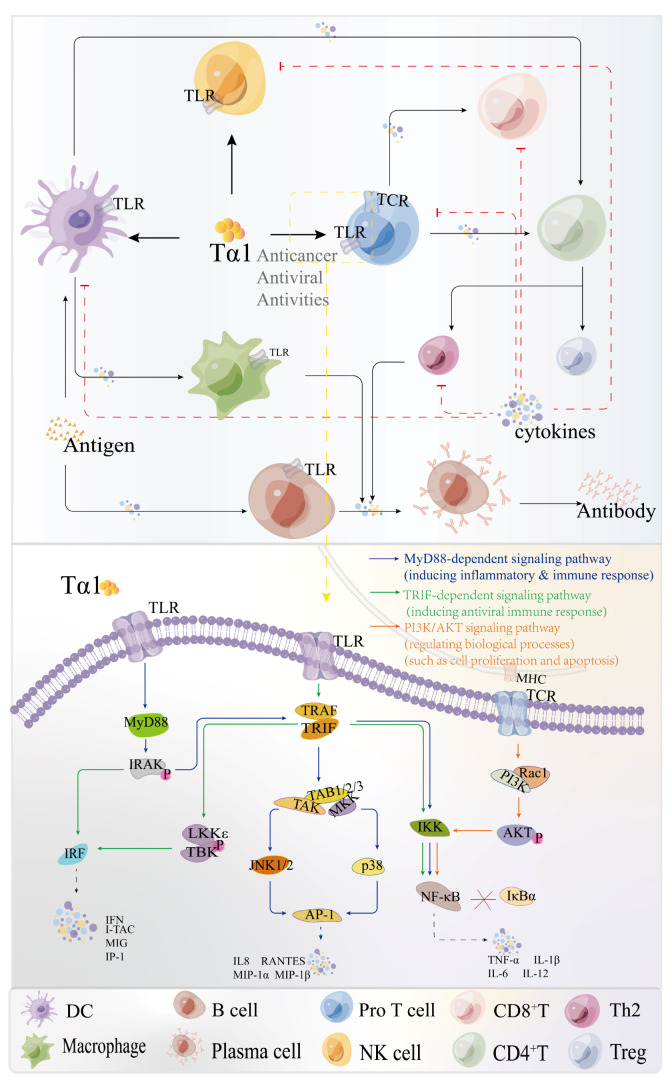
Immunomodulatory mechanism of Tα1 (DC: Dendritic cell; Pro T cell: Precursor T cell; NK cell: Natural killer cell; CD8^+^ T cell: Cytotoxic T cell; Treg cell: Regulatory T cell; Cytokines: IFN, IL-2, TNF, IL-10, et al.).

**Table 1 molecules-28-03539-t001:** Clinical research of Tα1 on COVID-19.

Dosage Regimen	Subjects with Tα1/Subjects	Main Results	Published Year
Tα1	269/317	▪ The level of NLR increased ▪ Patients with severe COVID-19 are not suitable for the treatment of glucocorticoid, immunoglobulin, thymosin, and ammonium glycyrrhizinate	2022 [76]
	306/1026	▪ Tα1 use in COVID-19 was associated with poor clinical outcomes ▪ Tα1 use at a later stage was significantly associated with a higher non-recovery rate than Tα1 use at an earlier stage ▪ The duration to use Tα1 had a less significant effect on recovery rate	2021 [74]
	126/275	▪ The use of Tα1 had no effect on promoting the recovery of CD4^+^ T cell and CD8^+^ T cell counts ▪ The severity of the disease and use of Tα1 may actually be related to a prolonged time of virus clearance	2021 [71]
	232/1388	▪ For non-severe patients with COVID-19, Tα1 can shorten viral RNA shedding duration and hospital stay but does not prevent the progression of COVID-19 and reduce the COVID-19-related mortality rate	2021 [68]
	27/47	▪ The reduction rate of complement C3 level and C-reactive protein level was higher than that of the control group (*p* = 0.01, 0.04), but it has no obvious positive effect on virus elimination ▪ Continuous monitoring of complement C3 levels and nucleic acid load are helpful for early assessment of discharge indications ▪ The immune system is closely related to the severity of COVID-19 and clinical outcomes	2021 [77]
	327/771	▪ The treatment of Tα1 was not associated with a difference in 28-day mortality in patients with COVID-19 after adjustment for baseline confounders ▪ Subgroup analysis and phenotype analysis did not show benefits in 28-day mortality with Tα1 therapy	2021 [75]
	78/127	▪ The male patients in the treatment group showed higher CRP and IL-6 levels after treatment, but the PCT level decreased significantly ▪ Gender differences may be a factor in sustaining the immunity respond of COVID-19 to Tα1	2020 [76]
	36/76	▪ Tα1 supplement significantly reduce mortality of severe COVID-19 patients ▪ COVID-19 patients with counts of CD8^+^ T cells or CD4^+^ T cells in circulation lower than 400/μL or 650/μL, respectively, gain more benefits from Tα1 ▪ Tα1 reverses T cell exhaustion and recovers immune reconstitution through promoting thymus output during SARS-CoV-2 infection	2020 [70]
	102/334	▪ Treatment with Tα1 can markedly decrease 28-day mortality and attenuate acute lung injury in critical-type COVID-19 patients	2020 [78]

## Data Availability

Not applicable.

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
