# Peer review of "Thymosin α1 and Its Role in Viral Infectious Diseases: The Mechanism and Clinical Application"

_molecules, 2023, doi:10.3390/molecules28083539_

Round 1

Reviewer 1 Report

The manuscript of Tao et al. is hardly understandable. There are a lot of language errors, unclear or not finished sentences, and missing or redundantly added words. More importantly, however, it contains a lot of factual errors. For instance:

a.       Line 66: in [16] there is no information about the nuclear localization of Tα1. On the contrary, [15] and [23] argue for a non-nuclear intracellular function of Tα1

b.       Line 74: it should be dimiristoylphospatidylcholine and dimiristoylphosphatic acid,  instead of dimethyl

c.       Line 83: “β-rotational conformations”? “Double rotations”? In the referenced [13] there were β-turns invoked, and they were detected by MD, not NMR.

d.       Line 86: in [15] conformation of Tα1 in a micelle consisting of DPC and SDS was described, not the conformation in pure SDS

e.       Line 87: “structural fracture between residues 1-9 and 14-25”. Nepravishta described structural breaks around residues 7 and 14 (Nepravishta et al, 2015). Regions 1-6 and 15-26 were described as α-helices.

f.        Line 115: “inserted […] through the free N-terminal region” Tα1 is acetylated at its N-terminus; there is no information in the referenced [15] about the free N-terminus.

g.       Line 199: in [44] there is no information that “transformation of Tα1 into cytotoxic T lymphocytes” is possible

h.       Line 300: [66] states that Tα1 treated patients had a higher number of lymphocytes than patients without treatment, not that “the potential mechanism of action of Tα1 therapy is often interpreted as enhancing the immune response by restoring immune-mediated lung injury and reducing lymphocytes in the immune system”

In some places, references to the literature are incorrect because the information given in the manuscript is not present in the source material. For instance:

a.       Line 91: in [18] there is no information about Tα1; the paper describes 310 helical structure

b.       Line 108: [21] describes the crystal structure of RCsB transcription regulator. The paper is not related to Tα1.

c.       Line 161: [33] underscores that Tα1 alone showed no efficacy against HCV chronic infection. Encouraging effects were observed only by combining Tα1 and IFN treatment.  

d.       Line 308: [68] describes baricitinib, not Tα1

e.       Line 310: [69] describes Covax-19 vaccine, not Tα1

The text is poorly structured, sometimes giving contradictory information without comment, for instance, „Tα1 can promote […] the expression of activation markers but not influence IL-12 production” (line 188) versus “it can regulate the immune function of the body by […] regulating the release of inflammatory cytokines such as IL-12” (line 196).

References are missing in several places:

a.       Line 163: “It can also inhibit the virus replication and virus protein expression by increasing the content of glutathione in cells.”

b.       Line 252: “enhancing T cell function; Increase the phagocytic activity of dendritic cells and cytotoxicity of NK cells; Activate Toll-like receptors, activate dendritic cells and macrophages, and start MAPK, Jak, NF-κB and other signaling pathways.”

c.       Line 316: “For patients whose blood CD8+T cells or CD4+T cells are lower than 400/μL or 650/μL, respectively, Tα1 can effectively increase the number of T cells in the blood of patients with severe lymphopenia.”

Last but not least:

-        The Figures are not legible

-        At the beginning of chapter 2.2, the text from the template has been left

-        There are many errors and inaccuracies in the text, for instance:

Line 76: What does it mean “Tα1 can interact with other natural proteins to form ordered proteins”?

Line 73: Tα1 is a peptide, not a protein

line 74: What does it mean “they usually show an inherently disordered protein”

line 104: “differences between the conformation of Tα1 and SDS micelles” in the caption of the Figure presenting two superimposed structures of Tα1

line 112: “negatively charged target membrane region exposed to phosphatidylserine” while the membrane is negatively charged due to phosphatidylserine exposition

line 192: “in the in the presence of CD4 on MHC class + DCs and enhance the up-regulated levels of mature markers induced by TNF- α , but also suppress the up-regulated IL-12 production.” In the presence of TNFα, Tα1 promoted the expression of CD4 on MHC class+ DCs

Line 279: it should be delayed immune reconstitution, not reconstruction

Table 2: 51.7% vs. 3.9% does not result in a 17.8% difference

Author Response

Response to Reviewer Comments (molecules-2289861)

Dear Sir/Madam,

Thank you very much for your letter, and we would like to express our sincere gratitude for the valuable chance you offered to re-submit our manuscript. It is highly appreciated for the positive and constructive comments and suggestions of you and the reviewers to our manuscript entitled “Thymosin α1 and its role in viral infectious diseases: the mechanism and clinical application” (Manuscript Number: molecules-2289861). Based on the comments, we have made extensive modifications to the original manuscript. Here, we attached the revised manuscript in the format of word for your approval, the revised part was marked with red and the deleted parts are also marked with a strikethrough in the manuscript. The response to every comment from the reviewers was listed by item as followed.

Best regards,

Jianli Gao

Ph D, Professor in Immunopharmacology,

ORCID: 0000-0002-5977-0021

Chief Editor of Evidence-Based Complementary and Alternative Medicine

Managing Editor of Clinical Complementary Medicine and Pharmacology

School of Pharmaceutical Sciences, Zhejiang Chinese Medical University, Hangzhou, China

Tel: 86-571-61768515 

Point 1: The manuscript of Tao et al. is hardly understandable. There are a lot of language errors, unclear or not finished sentences, and missing or redundantly added words.

Response 1: Thank you for your suggestion. We have carefully revised the manuscript. The revised part was marked with red and the deleted parts are also marked with strikethrough in the manuscript.

Point 2: Line 66: in [16] there is no information about the nuclear localization of Tα1. On the contrary, [15] and [23] argue for a non-nuclear intracellular function of Tα1

Response 2: Thank you for your advice. We apologize for our mistakes. After consulting the literature, we revised this part of the content in the revised draft. The revised content is as follows: 

In “New Line 74”: “(5) it may be involved in the regulation of the cell cycle [16];”

Point 3: Line 74: it should be dimiristoylphospatidylcholine and dimiristoylphosphatic acid,  instead of dimethyl

Response 3: Thank you for your advice. We apologize for our mistakes. After consulting the literature, we revised this part of the content in the revised draft. The revised content is as follows:

In “New Line 80”: “In the monolayer vesicle of dimiristoylphosphatidylcholine and dimiristoylphosphatidic acid (10:1), it showed a partially structured conformation [17].”

Point 4: Line 83: “β-rotational conformations”? “Double rotations”? In the referenced [13] there were β-turns invoked, and they were detected by MD, not NMR.

Response 4: Thank you for your advice. We apologize for our mistakes. After consulting the literature, we revised this part of the content in the revised draft. The revised content is as follows:

In “New Line 85”: “A structured conformation of the peptide was observed through restrained molecular dynamic simulations with an explicit solvent containing 40% TFE/60% TIP3P water (v/v) , with two stable regions identified: an alpha-helix region spanning residues 14 to 26, and two double β-turns in the N-terminal twelve residues, which form a distorted helical structure (Fig. 1a) [13].”

Point 5: Line 86: in [15] conformation of Tα1 in a micelle consisting of DPC and SDS was described, not the conformation in pure SDS.

Response 5: Thank you for your advice. After reviewing the literature, we revised this part in the revised draft and re-inserted the literature. The revision content as follows:

In “New Line 94”: “In contrast to TFE mixed solvents, Tα1 in sodium dodecyl sulfate (SDS) displays a spiral folded conformation (Fig. 1b) [18].”

The new DOI: https://doi.org/10.1517/14712598.2015.1009034

Point 6: Line 87: “structural fracture between residues 1-9 and 14-25”. Nepravishta described structural breaks around residues 7 and 14 (Nepravishta et al, 2015). Regions 1-6 and 15-26 were described as α-helices.

Response 6: Thank you for your valuable advice. I carefully read the literature and found that the content introduced in the literature I quoted was really "structural fracture between residues 1-9 and 14-25". It is worth noting that the document I quoted is the same author as the document you provided, and the year of the document is the same. But because there is only a small amount of information, I can't accurately judge whether the literature we are talking about belongs to the same article. If there are new clues to prove that the documents we refer to are not the same, please criticize and correct me, and I will definitely revise them carefully.

In “New Line 95” and the literature link in my article :

“This conformation is characterized by a structural fracture between residues 1 - 9 and 14 - 25, with the acetylated N-terminal residues 1 - 5 often inserted into the hydrophobic region of the micelle. The investigation also revealed that the folded conformation of Tα1 in SDS closely resembled that in phospholipid vesicles, taking on a 310 helical structure [18].”

The DOI: https://doi.org/10.1517/14712598.2015.1009034

Point 7: Line 115: “inserted […] through the free N-terminal region” Tα1 is acetylated at its N-terminus; there is no information in the referenced [15] about the free N-terminus.

Response 7: We sincerely thank the reviewers for their careful reading. We examined the literature carefully. For the sake of rigor, we have removed the word "free" in the revised draft we resubmitted. In addition, we have added more references about "N-terminal insertion into cell membrane" in this section. The revision content as follows:

In “New Line 123”: “Subsequently, the N-terminal region of Tα1 can insert into the hydrophobic region of the cell membrane, triggering a cascade reaction of biological signals (Fig. 2) [15; 20].”

The DOI: https://doi.org/10.1021/acs.biochem.5b01345

Point 8: Line 199: in [44] there is no information that “transformation of Tα1 into cytotoxic T lymphocytes” is possible

Response 8: We sincerely thank you for your valuable advice. We have carefully checked the literature and added more references about "transformation of Tα1 into cytotoxic T lymphocytes" in the revised draft. It is worth noting that Tα1 is a polypeptide secreted by thymocytes, which plays an important role in immune response. Tα1 can enhance the function of immune cells, including activating T cells and enhancing the activity of cytotoxic T lymphocytes. In addition, Tα1 can also enhance the activity of macrophages and natural killer cells, thus improving the immune ability of the body. Cytotoxic T lymphocytes are lymphocytes with the ability to kill tumor cells and infected cells. They are differentiated from T cells, and Tα1 can promote the differentiation of T cells into cytotoxic T lymphocytes, thus improving the immunity of the body, which has potential application value in the treatment of tumors and viral infections.

The revision content as follows:

In “New Line 218”: “Upon binding to TLR receptors located on the surface of precursor T cells, Tα1 promotes their differentiation into cytotoxic T lymphocytes (CD8+ T cells, CTL) [25; 26; 43].” 

The DOI: https://doi.org/10.1016/bs.vh.2016.04.003; https://doi.org/10.1111/j.1749-6632.2012.06697.x; https://doi.org/10.1196/annals.1415.032

Point 9: Line 300: [66] states that Tα1 treated patients had a higher number of lymphocytes than patients without treatment, not that “the potential mechanism of action of Tα1 therapy is often interpreted as enhancing the immune response by restoring immune-mediated lung injury and reducing lymphocytes in the immune system”

Response 9: Thank you for your valuable advice. We reviewed the relevant literature. In the study of Matteucci C(https://doi.org/10.1093/ofid/ofaa588 ), it is indicated that Tα1 can reduce the expression of cytokines in peripheral blood and specifically inhibit the activation of lymphocytes in T lymphocyte subsets of COVID-19 patients. But there are some mistakes in our expression. Therefore, we have revised this part.

The revision content as follows:

In “New Line 299”: “In addition, COVID-19 infection can cause changes in antigen presentation function or tissue resident immune cell pool, leading to the inability of the body to produce timely and effective antiviral response, thus reducing virus clearance rate and causing tissue damage [63; 64]”

The DOI: https://doi.org/10.1126/science.abm8108; https://doi.org/10.1093/ofid/ofaa588

Point 10: In some places, references to the literature are incorrect because the information given in the manuscript is not present in the source material. Line 91: in [18] there is no information about Tα1; the paper describes 310 helical structure

Response 10: Thank you for your advice. We apologize for our mistakes. After consulting the literature, we revised this part of the content in the revised draft. 

The revision content as follows:

In “New Line 98”:The investigation also revealed that the folded conformation of Tα1 in SDS closely resembled that in phospholipid vesicles, taking on a 310 helical structure [18]

The DOI: https://doi.org/10.1517/14712598.2015.1009034

Point 11: Line 108: [21] describes the crystal structure of RCsB transcription regulator. The paper is not related to Tα1.

Response 11: Thank you for your advice. We apologize for our mistakes. After consulting the literature, we revised this part of the content in the revised draft. 

The revision content as follows:

In “New Line 98”: “Despite the pleiotropic effects of Tα1 on immune regulation, the absence of specific receptors remains one of the primary factors contributing to the inefficacy of Tα1 therapy. However, recent research has demonstrated that the C-terminal region (residues 11-20) of Tα1, characterized by the amino acid sequence "LKEKK," may be capable of binding to human serum albumin (HSA) [20]”

The DOI: https://doi.org/10.1021/acs.biochem.5b01345 

Point 12: Line 161: [33] underscores that Tα1 alone showed no efficacy against HCV chronic infection. Encouraging effects were observed only by combining Tα1 and IFN treatment.  

Response 12: Thank you for your advice. We apologize for our mistakes. After consulting the literature, we revised this part of the content in the revised draft.

The revision content as follows:

In “New Line 171”: Tα1 upregulates the expression of major histocompatibility complex (MHC) antigens and B-2 microglobulin on the cell surface, resulting in an elevated expression level of virus-specific antigens and a consequent decrease in virus replication [27-30].

The DOI: https://doi.org/10.1080/14712598.2018.1451511 

Point 13: Line 308: [68] describes baricitinib, not Tα1

Response 13: Thank you for your advice. We apologize for our mistakes. After consulting the literature, we revised this part of the content in the revised draft.

The revision content as follows:

In “New Line 323”: Studies have shown that Tα1 strongly binds to angiotensin converting enzyme (ACE) in a very low concentration range, thus effectively down-regulating the expression of ACE 2, thus damaging the synthesis of angiotensin in human lung epithelial cells [70]

The DOI: https://doi.org/10.31083/j.fbl2702048 

Point 14: Line 310: [69] describes Covax-19 vaccine, not Tα1

Response 14: Thank you for your advice. We apologize for our mistakes. After consulting the literature, we revised this part of the content in the revised draft.

The revision content as follows:

In “New Line 326 ”: These results indicate that Tα1 may prevent COVID-19 by reducing the expression of ACE 2 in human lung epithelial cells [71]

The DOI: https://doi.org/10.1016/j.bioorg.2019.04.003 

Point 15: The text is poorly structured, sometimes giving contradictory information without comment, for instance, „Tα1 can promote […] the expression of activation markers but not influence IL-12 production” (line 188) versus “it can regulate the immune function of the body by […] regulating the release of inflammatory cytokines such as IL-12” (line 196).

Response 15: Thank you for your opinion. It may be difficult to read this part of the content because of our low quality expression. Therefore, we reviewed the relevant literature and revised this part in the revised draft. The revision content as follows:

In “New Line 215”: Furthermore, Tα1 can affect the immune function by inducing and regulating the maturation of T-cell and NK cells, activating lymphocytes, and regulating the secretion of inflammatory cytokines such as IL-2, IL-4, IFN-γ, TNF-α, et al [38; 41; 42]. Upon binding to TLR receptors located on the surface of precursor T cells, Tα1 promotes their differentiation into cytotoxic T lymphocytes (CD8+ T cells, CTL) [25; 26; 43].

The DOI: https://doi.org/10.1111/j.1749-6632.2012.06697.x ; https://doi.org/10.1016/bs.vh.2016.04.003 ; https://doi.org/10.1517/14712598.2015.1044895 ; https://doi.org/10.1111/j.1749-6632.2010.05465.x ; https://doi.org/10.12669/pjms.38.1.4419  https://doi.org/10.1196/annals.1415.032 ;

Point 16: References are missing in several places:

  1. Line 163: “It can also inhibit the virus replication and virus protein expression by increasing the content of glutathione in cells.”

Response 16(a): Thank you for your advice. We apologize for our mistakes. After consulting the literature, we revised this part of the content in the revised draft.

The revision content as follows:

In “New Line 175”: At the same time, one study has found that there is a negative correlation between glutathione content and influenza virus replication, indicating that glutathione may be an important factor in inhibiting influenza virus replication [32].

The DOI: https://doi.org/10.1165/rcmb.2021-0372OC 

  1. Line 252: “enhancing T cell function; Increase the phagocytic activity of dendritic cells and cytotoxicity of NK cells; Activate Toll-like receptors, activate dendritic cells and macrophages, and start MAPK, Jak, NF-κB and other signaling pathways.”

Response 16(b): Thank you for your valuable advice. However, it is worth noting that this part is our concluding remarks on part of the part of “2.4 The immunomodulatory mechanism of Tα1”. Relevant literature has been cited in the part of 2.4. Therefore, we don't repeat the references in this paragraph.

  1. Line 316: “For patients whose blood CD8+T cells or CD4+T cells are lower than 400/μL or 650/μL, respectively, Tα1 can effectively increase the number of T cells in the blood of patients with severe lymphopenia.”

Response 16(c): Thank you for your valuable advice. We checked this part and re-inserted new literature to support this conclusion.

The revision content as follows:

In “New Line 314”: For patients whose CD8+T cells or CD4+T cells are lower than 400/μL or 650/μL, respectively, Tα1 can effectively increase the number of T cells in the blood of patients with severe lymphopenia [68].

The DOI: https://doi.org/10.3389/fimmu.2021.568789 

Point 17: The Figures are not legible

Response 17: We apologize for our low-clarity diagram. We exported a higher resolution (1200 dpi) diagram according to your suggestion and inserted it into the revised manuscript. Of noted that the diagram in the manuscript would be compressed, and the clarity may be insufficient. So we provide the original diagram in the Submission system.

Point 18: At the beginning of chapter 2.2, the text from the template has been left

Response 18: We are very sorry for our careless mistakes. This part has been deleted and corrected in our revised draft.

 In “New Line 77”: Under natural conditions, Tα1 is a short, highly charged, and inherently disordered protein. At neutral pH and 37℃, Tα1 typically exhibits intrinsic disorder, meaning that it does not have a stable, defined structure [14].

Point 19: Line 76: What does it mean “Tα1 can interact with other natural proteins to form ordered proteins”?

Response 19: Thank you very much for your valuable advice. We are very sorry for the reading trouble caused by the expression problem of the original text. It is worth noting that in this paper, we mentioned that Tα1 will be in a disordered state in the natural state, and it will present the structure in Figure 1 under certain conditions. However, in the revised draft, we corrected the expression of this part.

The revision content as follows:

In “New Line 82”: Under low pH condition, Tα1 has the capacity to form ordered protein complexes through interaction with other naturally occurring proteins.

Point 20: Line 73: Tα1 is a peptide, not a protein

Response 20: Thank you very much for your valuable advice. Yes, Tα1 is a polypeptide. But we need to explain that Tα1 is a peptide chain composed of 28 amino acids, which makes it a kind of protein. He is also called protein in the document https://doi.org/10.1016/bs.vh.2016.04.009 . 

Point 21: line 74: What does it mean “they usually show an inherently disordered protein”

Response 21: Thank you very much for your valuable advice. Under normal conditions of neutral pH and body temperature, when Tα1 is dissolved in water, the protein is considered to be inherently disordered. This means that Tα1 has no fixed and stable structure or shape, but exists as a flexible and dynamic molecule, lacking a specific three-dimensional conformation. This is a characteristic of protein's internal disorder. Although Tα1 lacks definite structure, it can play an important role in various cellular processes. In order to make it easier to understand, this part of the statement has been corrected in our revised draft. The revision content as follows:

In “New Line 78”: At neutral pH and 37℃, Tα1 typically exhibits intrinsic disorder, meaning that it does not have a stable, defined structure [14].

Point 22: line 104: “differences between the conformation of Tα1 and SDS micelles” in the caption of the Figure presenting two superimposed structures of Tα1

Response 22: Thank you for your valuable advice. We are sorry for our previous writing mistakes. We carefully examined this part and revised it in the revised draft.

The revision content as follows:

In “New Line 109”: Tertiary structure of Tα1 in different solvents (analyzed and processed by Pymol software). a. Spatial conformation of Tα1 in a TFE mixed solvent (PDB. 2L9I); b. Spatial conformation of Tα1 in an SDS micellar solvent (PDB. 2MNQ); c. Details of the structural differences between Tα1 (in green) and SDS micelles (in brown) in a mixed solvent (obtained through superposition comparison of protein structures using Pymol software).

Point 23: line 112: “negatively charged target membrane region exposed to phosphatidylserine” while the membrane is negatively charged due to phosphatidylserine exposition

Response 23: Thank you for your valuable advice. It may be difficult to understand this part because of our mistakes in expression. We reviewed the relevant literature and revised this part.

The revision content as follows:

In “New Line 121”: In the presence of a negatively charged target membrane region that exposes phosphatidylserine (PS), the C-terminal amino acid sequence of Tα1 can be transported to the vicinity of the target membrane by HSA.

Point 24: line 192: “in the in the presence of CD4 on MHC class Ⅱ + DCs and enhance the up-regulated levels of mature markers induced by TNF- α , but also suppress the up-regulated IL-12 production.” In the presence of TNFα, Tα1 promoted the expression of CD4 on MHC class+ DCs

Response 24: Thank you for your valuable comments. Perhaps our mistakes in expression have made this part ambiguous. We reviewed the relevant literature and revised this part.

The revision content as follows:

In “New Line 207”: Huang et al. demonstrated that, during the maturation of bone marrow-derived DCs (BMDC), Tα1 promoted the differentiation of CD4-expressing dendritic cells (DCs) and the expression of activation markers, but without affecting the production of IL-12, as well as the T cell-stimulatory capacity of DCs in the absence of TNF-α [36]. However, in the presence of TNF-α, Tα1 has been shown to not only increase the expression of CD4 on MHC class Ⅱ+ DCs and enhance the up-regulation of mature markers induced by TNF-α, but also suppress the up-regulation of IL-12 production. These effects were particularly evident at the pharmacological concentrations of Tα1.

Point 25: Line 279: it should be delayed immune reconstitution, not reconstruction

Response 25: Thank you for your valuable comments. Perhaps our mistakes in expression have made this part ambiguous. We reviewed the relevant literature and revised this part.

The revision content as follows:

In “New Line 404”: Acquired immunodeficiency syndrome (AIDS) is an immunodeficiency disease caused by human immunodeficiency virus (HIV) infection. It mainly targets at cells that can express CD4 in the immune system, such as DC, macrophages, and CD4+ T cells [86]. A study shows that the combined therapy of Tα1, zidovudine and IFN-α can effectively improve the level of CD4+ T cells in patients with AIDS [87]. However, there is not enough evidence to show that Tα1 can effectively control HIV infection as a monotherapy, so it is necessary to further verify its efficacy and safety.

Point 26: Table 2: 51.7% vs. 3.9% does not result in a 17.8% difference

Response 26: We sincerely thank you for your valuable advice. We carefully checked the references, and the original content was 33.9%, not 3.9%. We are sorry for our carelessness. After checking the original text, we summarized this part again.

The revision content as follows:

In “New Line 411”:  the CTU group exhibited a better performance with respect to organ failure scores; significant improvements in CD4(+)CD8(+) count after initiation of treatment; the balance between proinflammatory mediators and anti-inflammatory cytokines was better modulated

We hope that these amendments can solve the problems raised by reviewers and meet the publishing standards. Thank you for your time and consideration.

Reviewer 2 Report

This paper reviewed the characters of Thymosin α1 as an peptide hormone secreted by thymic stromal cells, and Tα1 plays important roles with immunoregulatory ability and as an clinical treatment in viral infectious diseases. The topic meets this journal’s requirement but some of the details are just listed out without neatly organized. Figure.3 is not very well organized (a lot of factors listed but not focused on Tα1).

Please revise this paper before accept.

Author Response

Thank you very much for your valuable advice. We carefully examined and revised the manuscript without changing the content and structure of the article. In response to your detailed questions, our additions and deletions are not listed here, but they are highlighted in red in the revised draft we resubmitted. In response to your question about Figure 3, we rearranged the contents of the picture.

Reviewer 3 Report

The work submitted for review requires significant revision in terms of grammatically correct spelling in English. A full revision by a native English speaker is desirable. At the moment, despite the potential usefulness and interest of this scientific review for researchers involved in studies of Thymosin α1 (Tα1) and other immunostimulatory peptides, it is difficult to read the material due to the large number of incorrectly composed and poorly written sentences. The following texts are a good example: Lines 61-65: “In addition, there are some basic characteristics of Tα1: (1) With 28 amino[12] acid residues and a relative molecular weight of 3108 Da; (2) Highly acidic with an isoelectric point of 4.2; (3) The N-terminal has an acetylation structure, and without disulfide bond or glycosylation structure[13-15].” Lines 72-74: “Under natural conditions, Tα1 is a short, highly charged and basically unstructured protein. At neutral pH or 37, they usually show an inherently disordered protein.” Text is poorly written, which makes it hard to understand from the first read. Often the sentences do not agree with each other, as in the second example, where the word "they" appears out of nowhere.  

Author Response

Response to Reviewer Comments (molecules-2289861)

Dear Sir/Madam,

Thank you very much for your letter, and we would like to express our sincere gratitude for the valuable chance you offered to re-submit our manuscript. It is highly appreciated for the positive and constructive comments and suggestions of you and the reviewers to our manuscript entitled “Thymosin α1 and its role in viral infectious diseases: the mechanism and clinical application” (Manuscript Number: molecules-2289861). Based on the comments, we have made extensive modifications to the original manuscript. Here, we attached the revised manuscript in the format of word for your approval, the revised part was marked with red and the deleted parts are also marked with a strikethrough in the manuscript. The response to every comment from the reviewers was listed by item as followed.

Best regards,

Jianli Gao

Ph D, Professor in Immunopharmacology,

ORCID: 0000-0002-5977-0021

Chief Editor of Evidence-Based Complementary and Alternative Medicine

Managing Editor of Clinical Complementary Medicine and Pharmacology

School of Pharmaceutical Sciences, Zhejiang Chinese Medical University, Hangzhou, China

Tel: 86-571-61768515 

Point 1: The work submitted for review requires significant revision in terms of grammatically correct spelling in English. A full revision by a native English speaker is desirable. At the moment, despite the potential usefulness and interest of this scientific review for researchers involved in studies of Thymosin α1 (Tα1) and other immunostimulatory peptides, it is difficult to read the material due to the large number of incorrectly composed and poorly written sentences. The following texts are a good example: Lines 61-65: “In addition, there are some basic characteristics of Tα1: (1) With 28 amino[12] acid residues and a relative molecular weight of 3108 Da; (2) Highly acidic with an isoelectric point of 4.2; (3) The N-terminal has an acetylation structure, and without disulfide bond or glycosylation structure[13-15].” Lines 72-74: “Under natural conditions, Tα1 is a short, highly charged and basically unstructured protein. At neutral pH or 37℃, they usually show an inherently disordered protein.” Text is poorly written, which makes it hard to understand from the first read. Often the sentences do not agree with each other, as in the second example, where the word "they" appears out of nowhere.

Response 1: Thank you very much for your valuable advice. In response to your poor grammar and writing. We carefully examined and revised the manuscript without changing the content and structure of the article. The additions and deletions we made are not listed here, but they are highlighted in red in the revised draft we resubmitted.

In view of your poor handwriting, we have revised it in the revised draft (some problems and corrections are as follows):

  1. Lines 61-65: “In addition, there are some basic characteristics of Tα1: (1) With 28 amino[12] acid residues and a relative molecular weight of 3108 Da; (2) Highly acidic with an isoelectric point of 4.2; (3) The N-terminal has an acetylation structure, and without disulfide bond or glycosylation structure[13-15].”

In “New Line 69”: In addition, Tα1 has several notable characteristics: (1) it consists of 28 amino acid residues and has a relative molecular weight of 3108 Da [12]; (2) it is highly acidic with an isoelectric point of 4.2; (3) the N-terminal is acetylated and lacks disulfide bonds or glycosylation structures [13-15]. (4) the entire polypeptide molecule has six amino acid repeats (Ala-Ala, Ser-Ser, Thr-Thr, Lys-Lys, Val-Val, Glu-Glu); (5) it may be involved in the regulation of the cell cycle [16]; (6) it can participate in transcription and/or DNA replication.

  1. Lines 72-74: “Under natural conditions, Tα1 is a short, highly charged and basically unstructured protein. At neutral pH or 37℃, they usually show an inherently disordered protein.”

In “New Line 77”: Under natural conditions, Tα1 is a short, highly charged, and inherently disordered protein. At neutral pH and 37℃, Tα1 typically exhibits intrinsic disorder, meaning that it does not have a stable, defined structure [14]

We hope that these amendments can solve the problems raised by reviewers and meet the publishing standards. Thank you for your time and consideration.

Round 2

Reviewer 1 Report

The authors thoroughly modified the text, responding to comments made by reviewers. The work is now a clearer and more understandable presentation of information reported in the literature. Nevertheless, a few places still need improvement, namely:

Lines 37 -41: Two identical sentences appear consecutively in the text; they differ only in the cited reference.

The identification and isolation of thymosin date back to 1966 when Goldstein and colleagues first discovered a lymphocyte production factor in the calf thymus, subsequently named thymosin [5]. The identification and isolation of thymosin date back to 1966 when Goldstein and colleagues first discovered a lymphocyte production factor in the calf thymus, subsequently named thymosin [6].

Lines 87-91: Almost identical information was provided in two consecutive sentences:

with two stable regions identified: an alpha-helix region spanning residues 14 to 26, and two double β-turns in the N-terminal twelve residues, which form a distorted helical structure (Fig. 1a) [13]. Stable α-helical fragments were observed in the C-terminal region between residues 14-26, and stable twisted helical structures were observed in the N-terminal region between residues 1-12”.

Line 99: It should be 310 not 310 helical structure

Line 166: It should be “ […] and impacts” not “and to impact”

Lines 269-272: There are still some problems with punctuation in this fragment:

·         It is not a full sentence: On the other hand, after the virus enters the body.

·         Why are quotation marks present in Enhancing T-cell function; Activate dendritic cells and macrophages, increase the phagocytic activity of dendritic cells and cytotoxicity of NK cells; Activate Toll-like receptors, and start MAPK, Jak, NF-κB and other signaling pathways. If any paper is quoted here it should be referenced.

Lines 327-328: There is no information about COVID-19 in [71]. The paper describes in general inhibitory effect of Tα1 on angiotensin-converting enzyme (ACE) and does not refer to human lung epithelial cells. Possibly, references [70] and [71[ were swapped. 

“N-terminal” is an adjective; in several places it was used by the author as a noun (line 72, 92, 103)

Author Response

Response to Reviewer Comments 

 Dear Sir/Madam,

Thank you very much for your letter, and we would like to express our sincere gratitude for the valuable chance you offered to re-submit our manuscript. It is highly appreciated for the positive and constructive comments and suggestions of you and the reviewers to our manuscript entitled “Thymosin α1 and its role in viral infectious diseases: the mechanism and clinical application” (Manuscript Number: molecules-2289861). Based on the comments, we have made extensive modifications to the original manuscript. Here, we attached the revised manuscript in the format of word for your approval, the revised part was marked with red and the deleted parts are also marked with a strike through in the manuscript. The response to every comment from the reviewers was listed by item as followed.

Best regards,

Jianli Gao

Ph D, Professor in Immunopharmacology,

ORCID: 0000-0002-5977-0021

Chief Editor of Evidence-Based Complementary and Alternative Medicine

Managing Editor of Clinical Complementary Medicine and Pharmacology

School of Pharmaceutical Sciences, Zhejiang Chinese Medical University, Hangzhou, China

Tel: 86-571-61768515

Point 1: Lines 37 -41: Two identical sentences appear consecutively in the text; they differ only in the cited reference.

The identification and isolation of thymosin date back to 1966 when Goldstein and colleagues first discovered a lymphocyte production factor in the calf thymus, subsequently named thymosin [5]. The identification and isolation of thymosin date back to 1966 when Goldstein and colleagues first discovered a lymphocyte production factor in the calf thymus, subsequently named thymosin [6].

Response 1: Thank you very much for your advice. I carefully checked the literature and made the following corrections to this repetition:

New Line38: “Thymosin was discovered and isolated in 1966 by Goldstein and colleagues, who found a lymphocyte production factor in the calf thymus that was later termed thymosin [6; 7]. A protein combination including three short peptides (molecular weight less than 4000) known as thymosin Fraction 5 (TF 5) was produced after separation and purification [8].”

Point 2: Lines 87-91: Almost identical information was provided in two consecutive sentences:

“with two stable regions identified: an alpha-helix region spanning residues 14 to 26, and two double β-turns in the N-terminal twelve residues, which form a distorted helical structure (Fig. 1a) [13]. Stable α-helical fragments were observed in the C-terminal region between residues 14-26, and stable twisted helical structures were observed in the N-terminal region between residues 1-12”.

Response 2: Thank you very much for your advice. We carefully checked the original text and deleted this repetition:

New Line85: “A structured conformation of the peptide was observed through restrained molecular dynamic simulations with an explicit solvent comprising 40% TFE/60% TIP3P water (v/v) , with two stable regions identified: an alpha-helix region spanning residues 14 to 26, and two double β-turns in the N-terminal twelve residues of Tα1, which form a distorted helical structure (Fig. 1a) [15]. Additionally, two β-rotational conformations were detected at the N-terminal of Tα1, namely (I, I+1)-double rotations of the residues ASP2-ASP6 and (I, I+2)-double rotations of the residues Thr7-Thr12.”

Point 3: Line 99: It should be 310 not 310 helical structure

Response 3: Thank you very much for your advice. We are sorry for this mistake. We carefully checked the literature and corrected this mistake:

New Line95: “The investigation also revealed that the folded conformation of Tα1 in SDS closely resembled that in phospholipid vesicles, taking on a 310 helical structure [21].”

Point 4: Line 166: It should be “ […] and impacts” not “and to impact”

Response 4: Thank you very much for your feedback and suggestions. We carefully checked and corrected the grammar in the manuscript to ensure their accuracy. The error here has been corrected as follows:

New Line168: “Tα1 exerts an immune modulatory activity on T cell and NK cells, and impacts the functions of mature lymphocytes, such as stimulating cytokine production and cytotoxic T-lymphocyte-mediated cytotoxic responses [29].”

Point 5: Lines 269-272: There are still some problems with punctuation in this fragment:

It is not a full sentence: On the other hand, after the virus enters the body.

Response 5: Thank you very much for your advice. We carefully checked the original text and made the following corrections here to facilitate reading. At the same time, we also conducted a comprehensive inspection of the punctuation of the full text to ensure that our writing conforms to academic norms:

New Line274: “……; on the other hand, after the virus enters the body, Tα1 can treat viral diseases by enhancing T cell function, …...”

Point 6: Why are quotation marks present in “Enhancing T-cell function; Activate dendritic cells and macrophages, increase the phagocytic activity of dendritic cells and cytotoxicity of NK cells; Activate Toll-like receptors, and start MAPK, Jak, NF-κB and other signaling pathways ”. If any paper is quoted here it should be referenced.

Response 6: Thank you for your valuable advice. However, it is worth noting that this part is our

concluding remarks on part of the part of “2.4 The immunomodulatory mechanism of Tα1”. Relevant literature has been cited in the part of 2.4. Therefore, we don't repeat the references in this paragraph. But for the completeness of this part, we have updated the references of this part. The corrections are as follows:

New Line271: “To sum up, the antiviral effect of Tα1 can be summarized into two aspects: on the one hand, Tα1 can directly inhibit virus replication and viral protein expression by increasing the expression of cell surface-related antigens [19; 31-33]; on the other hand, after the virus enters the body, Tα1 can treat viral diseases by enhancing T cell function, activating dendritic cells and macrophages, increasing the phagocytic activity of dendritic cells and the cytotoxicity of NK cells, activating TLRs, and starting MAPK, Jak, NF-κB, and other signaling pathways [22; 51; 54; 55; 57-59]. “

New DOI: https://doi.org/10.1002/1521-4141(200003)30:3<778::aid-immu778>3.0.co;2-i; https://doi.org/10.1038/sj.embor.7400433; https://doi.org/10.1196/annals.1415.039; https://doi.org/10.1196/annals.1415.044; https://doi.org/10.1080/14712598.2018.1451511; https://doi.org/10.1196/annals.1415.025; https://doi.org/10.1038/nri1391; https://doi.org/10.1016/s1567-5769(01)00139-4; https://doi.org/10.1016/j.imlet.2007.04.007; https://doi.org/10.1093/intimm/dxm097; https://doi.org/10.1196/annals.1415.002

Point 7: Lines 327-328: There is no information about COVID-19 in [71]. The paper describes in general inhibitory effect of Tα1 on angiotensin-converting enzyme (ACE) and does not refer to human lung epithelial cells. Possibly, references [70] and [71] were swapped.

Response 7: Thank you very much for your advice. We carefully examined the original text and references, and corrected the order of the documents:

New Line323: “Tα1 significantly binds to angiotensin converting enzyme (ACE) at extremely low concentrations, effectively down-regulating the expression of ACE 2 and impairing the synthesis of angiotensin in human lung epithelial cells [72]. These results indicate that Tα1 may prevent COVID-19 by reducing the expression of ACE 2 in human lung epithelial cells [72; 73].”

DOI: [72] https://doi.org/10.31083/j.fbl2702048; [73] https://doi.org/10.1016/j.bioorg.2019.04.003

Point 8: “N-terminal” is an adjective; in several places it was used by the author as a noun (line 72, 92, 103)

Response 8: Thank you very much for your advice. We carefully checked the original text and changed " the N-terminal " to "the N-terminal of Tα1". The specific changes have been marked in red in the text.

New Line71: “(3) the N-terminal of Tα1 is acetylated, and……”

New Line88: “……β-turns in the N-terminal twelve residues of Tα1,……”

New Line90: “……at the N-terminal of Tα1, namely (I, I+1)-double ……”

New Line94: “……, with the acetylated N-terminal residues 1–5 of Tα1 often inserted into the hydrophobic……”

New Line102: “……and insert the N-terminal of Tα1 into the hydrophobic……”

New Line【123】: “The N-terminal region of Tα1 can then enter……”

Reviewer 2 Report

The authors revised the text and Figure.3 as well. Seems much better and meet the criteria and ready to be published.

Author Response

Response to Reviewer Comments

 Dear Sir/Madam,

Thank you very much for your letter, and we would like to express our sincere gratitude for the valuable chance you offered to re-submit our manuscript. It is highly appreciated for the positive and constructive comments and suggestions of you and the reviewers to our manuscript entitled “Thymosin α1 and its role in viral infectious diseases: the mechanism and clinical application” (Manuscript Number: molecules-2289861). Based on the comments, we have made extensive modifications to the original manuscript. Here, we attached the revised manuscript in the format of word for your approval, the revised part was marked with red and the deleted parts are also marked with a strike through in the manuscript. The response to every comment from the reviewers was listed by item as followed.

Best regards,

Jianli Gao

Ph D, Professor in Immunopharmacology,

ORCID: 0000-0002-5977-0021

Chief Editor of Evidence-Based Complementary and Alternative Medicine

Managing Editor of Clinical Complementary Medicine and Pharmacology

School of Pharmaceutical Sciences, Zhejiang Chinese Medical University, Hangzhou, China

Tel: 86-571-61768515

The authors revised the text and Figure.3 as well. Seems much better and meet the criteria and ready to be published.

Response: Thanks for your positive comments.

Reviewer 3 Report

Line 11 and line 277

“which is usually used as an immune enhancer in viral infectious diseases such as hepatitis B, HIV/AIDS, and sepsis”

“…a diverse range of viral infectious diseases, including sepsis, hepatitis B, among others.”

Sepsis is not an infectious disease, but a condition that is caused by an infection

Lines 37-40

“The identification and isolation of thymosin date back to 1966 when Goldstein and colleagues first discovered a lymphocyte production factor in the calf thymus, subsequently named thymosin [5]. The identification and isolation of thymosin date back to 1966 when Goldstein and colleagues first discovered a lymphocyte production factor in the calf thymus, subsequently named thymosin [6]”

The same information is repeated two times

Lines 63-74

“Tα1 (Generic drug name: thymalfasin; trade name: Zadaxin) is a bioactive peptide with 28 amino acid residues…”

“…(1) it consists of 28 amino acid residues and has a relative molecular weight of 3108 Da”

The same information is repeated

Lines 73-74

“It can participate in transcription and/or DNA replication”

This statement must be supported by a reference to literature

FIGURE 1

1) The amino acid labels in the figure are too small and of poor quality.

2) There should not be any dots after the PDB abbreviation. The right form is PDB XXXX or PDB ID XXXX, not PDB. XXXX

3) Pymol software should be written correctly (PyMOL)

Lines 153-155

“A pharmacokinetic study showed that after subcutaneous injection, the body absorbed well, and the serum level reached its peak at 1 - 2 h, with a half-life of less than 3 h”

It is not clear from the text what is the “serum level” and what is its “half-life”. This should be elaborated.

Lines 342-343

COVID-19 and Covid-19 are written differently.

Lines 417-418

“As a polypeptide, Tα1 possesses high selectivity and thus serves as a promising lead compound.”

This statement must be supported by a reference to literature

+ There are different spelling variants of the scientific terms in the text. For example, T-cell and T cell; B-cells and B cells; CD4+ T cells and CD4 T cells etc. Such negligence leads me to the conclusion that the author mindlessly copied passages from various articles and did not even bother to bring everything to a single form

There are several incorrect spellings of “et al.”. It should be written with a dot after the word “al”.

Unfortunately, the problem that it’s difficult to read this review because of the poor English grammar stays. The example are lines 345-350:

“In the first stage, the early stage of infection, the body usually does not show any other abnormalities except (for?) lymphopenia and neutropenia. Antiviral therapy is beneficial. In the second stage, the patient's inflammatory level began to rise (begins?), and antiviral therapy should be used cautiously and reasonably. In the third stage, the markers of patients' inflammatory level increase (does their titer increases?), which makes them prone to "cytokine storm" reaction”

Author Response

Response to Reviewer Comments

 Dear Sir/Madam,

Thank you very much for your letter, and we would like to express our sincere gratitude for the valuable chance you offered to re-submit our manuscript. It is highly appreciated for the positive and constructive comments and suggestions of you and the reviewers to our manuscript entitled “Thymosin α1 and its role in viral infectious diseases: the mechanism and clinical application” (Manuscript Number: molecules-2289861). Based on the comments, we have made extensive modifications to the original manuscript. Here, we attached the revised manuscript in the format of word for your approval, the revised part was marked with red and the deleted parts are also marked with a strike through in the manuscript. The response to every comment from the reviewers was listed by item as followed.

Best regards,

Jianli Gao

Ph D, Professor in Immunopharmacology,

ORCID: 0000-0002-5977-0021

Chief Editor of Evidence-Based Complementary and Alternative Medicine

Managing Editor of Clinical Complementary Medicine and Pharmacology

School of Pharmaceutical Sciences, Zhejiang Chinese Medical University, Hangzhou, China

Tel: 86-571-61768515

Point 1: Line 11 and line 277

“which is usually used as an immune enhancer in viral infectious diseases such as hepatitis B, HIV/AIDS, and sepsis”

“…a diverse range of viral infectious diseases, including sepsis, hepatitis B, among others.”

Sepsis is not an infectious disease, but a condition that is caused by an infection

Response 1: Thank you very much for your advice. We carefully revised and adjusted this part. The specific corrections are as follows:

New Line278: “3. The application of Tα1 in viral diseases and its complications.”

New Line372: “3.2. Other viral diseases and its complications.”

New Line404: “Virus infection often causes inflammatory reaction and affects the normal function of the immune system, which leads to the decline of the body's immunity and increases the risk of infection with bacteria and other pathogens. Sepsis is an organ dysfunction syndrome caused by the imbalance of human response to infection [88; 89]. Its main characteristic is systemic inflammatory response syndrome (SIRS) triggered by an infection [90]. According to Chen et al., the combined administration of Tα1 and Xuebijing has been shown to improve the therapy effect of severe pneumonia complicated with sepsis, improve the hemorheology condition of patients, and reduce the expression levels of serum CRP, TNF-α, IL-6, IL-8 and other inflammatory factors in patients [91]”

Point 2: Lines 37-40

“The identification and isolation of thymosin date back to 1966 when Goldstein and colleagues first discovered a lymphocyte production factor in the calf thymus, subsequently named thymosin [5]. The identification and isolation of thymosin date back to 1966 when Goldstein and colleagues first discovered a lymphocyte production factor in the calf thymus, subsequently named thymosin [6]”

The same information is repeated two times

Response 2: Thank you very much for your advice. We carefully examined the literature and made the following corrections to this repetition

New Line38: “Thymosin was discovered and isolated in 1966 by Goldstein and colleagues, who found a lymphocyte production factor in the calf thymus that was later termed thymosin [6; 7].”

Point 3: Lines 63-74

“Tα1 (Generic drug name: thymalfasin; trade name: Zadaxin) is a bioactive peptide with 28 amino acid residues…”

“…(1) it consists of 28 amino acid residues and has a relative molecular weight of 3108 Da”

The same information is repeated

Response 3: Thank you very much for your advice. We carefully checked the content of the original text and made some deletions to the repeated parts. The changes are as follows:

New Line70: “… (1) it has a relative molecular weight of 3108 Da [14]”

Point 4: Lines 73-74

“It can participate in transcription and/or DNA replication”

This statement must be supported by a reference to literature

Response 4: Thank you very much for your advice. We carefully examined the original text and literature. This part has been corrected and the references have been updated. The changes are as follows

New Line74: “(6) it can indirectly affect transcription and/or DNA replication [19].”

New DOI: https://doi.org/10.1002/1521-4141(200003)30:3<778::aid-immu778>3.0.co;2-i

Point 5: FIGURE 1

1) The amino acid labels in the figure are too small and of poor quality.

Response 5.1: Thank you very much for your advice. We made changes to the Figure 1. The specific changes have been marked in the manuscript.

2) There should not be any dots after the PDB abbreviation. The right form is PDB XXXX or PDB ID XXXX, not PDB. XXXX

Response 5.2: Thank you very much for your advice. We carefully examined the original text and made the following corrections to this part:

New Line110: “a. Spatial conformation of Tα1 in a TFE mixed solvent (PDB 2L9I); b. Spatial conformation of Tα1 in an SDS micellar solvent (PDB 2MNQ);”

3) Pymol software should be written correctly (PyMOL)

Response 5.3: Thank you very much for your advice. We carefully examined the original text and made the following corrections to this part:

New Line109: “Figure 1. Tertiary structure of Tα1 in different solvents (analyzed and processed by PyMOL software) ……. comparison of protein structures using PyMOL software).”

Point 6: Lines 153-155

“A pharmacokinetic study showed that after subcutaneous injection, the body absorbed well, and the serum level reached its peak at 1 - 2 h, with a half-life of less than 3 h”

It is not clear from the text what is the “serum level” and what is its “half-life”. This should be elaborated.

Response 6: Thank you very much for your advice. We carefully examined the original text and literature, made some corrections to this part and added new literature to support this paragraph.

New Line154: “Pharmacokinetic study showed that after subcutaneous injection, Tα1 is absorbed well in the body, and its peak blood drug concentration (Cmax, the highest blood drug concentration after administration) is reached at 1-2 hours, with a plasma half-life (t1/2, an estimate of the time it takes for the concentration or amount in the body of that drug to be reduced by exactly one-half (50%)) of less than 3 hours [27]”

New DOI: https://doi.org/10.1093/ajhp/58.10.886

Point 7: Lines 342-343

COVID-19 and Covid-19 are written differently.

Response 7: Thank you very much for your advice. We apologize for our carelessness. We carefully examined the original text and made the following corrections to this part:

New Line346: “Another reason is that the immune response mechanism caused by COVID-19 may be different from other infections.”

Point 8: Lines 417-418

“As a polypeptide, Tα1 possesses high selectivity and thus serves as a promising lead compound.”

This statement must be supported by a reference to literature

Response 8: Thank you very much for your feedback and suggestions. We have supplemented the references in this part

New Line426: “As a polypeptide, Tα1 possesses high selectivity and thus serves as a promising lead compound [100].”

New DOIhttps://doi.org/10.1007/s10989-022-10397-y

Point 9: There are different spelling variants of the scientific terms in the text. For example, T-cell and T cell; B-cells and B cells; CD4+ T cells and CD4 T cells etc. Such negligence leads me to the conclusion that the author mindlessly copied passages from various articles and did not even bother to bring everything to a single form.

Response 9: Thank you very much for your feedback and suggestions. We carefully check and correct all the scientific terms used in the paper to ensure their consistency and accuracy. Thank you again for your review and valuable feedback, and we will try our best to ensure that similar problems will not happen again. The specific changes have been marked in red in the text.

Point 10: There are several incorrect spellings of “et al.”. It should be written with a dot after the word “al”.

Response 10: Thank you very much for your feedback and suggestions. We carefully check and correct all the scientific terms used in the paper to ensure their consistency and accuracy. Thank you again for your review and valuable feedback, and we will try our best to ensure that similar problems will not happen again. The specific changes have been marked in red in the text.

Point 11: Unfortunately, the problem that it’s difficult to read this review because of the poor English grammar stays. The example are lines 345-350:

“In the first stage, the early stage of infection, the body usually does not show any other abnormalities except (for?) lymphopenia and neutropenia. Antiviral therapy is beneficial. In the second stage, the patient's inflammatory level began to rise (begins?), and antiviral therapy should be used cautiously and reasonably. In the third stage, the markers of patients' inflammatory level increase (does their titer increases?), which makes them prone to "cytokine storm" reaction”

Response 11: Thank you for your feedback. We apologize for any inconvenience caused by the poor English grammar in our review. We will make every effort to address the language issues in our manuscript to ensure that it is more readable and understandable for our readers. The specific changes are marked in red in the text. In view of the example, we have made the following changes:

New Line【348】: “Siddiqi et al. shown that the immunological response to the infection of COVID-19 is separated into three stages: Stage Ⅰ is an asymptomatic incubation phase, the body usually shows no further abnormalities except for lymphopenia and neutropenia. Antiviral treatment is advantageous. Stage Ⅱ is the non-severe symptomatic phase, the patient's inflammatory level began to rise, and antiviral medication should be taken with caution and reason. Stage Ⅲ is the severe respiratory symptomatic stage, patients' inflammatory markers such as IL-6, IL-8 and TNF-α rise, making them prone to a "cytokine storm" response.”